# Circulating Extracellular Vesicle MicroRNA as Diagnostic Biomarkers in Early Colorectal Cancer—A Review

**DOI:** 10.3390/cancers12010052

**Published:** 2019-12-23

**Authors:** Brendan J. Desmond, Elizabeth R. Dennett, Kirsty M. Danielson

**Affiliations:** 1Department of Surgery & Anaesthesia, University of Otago Wellington, 23 Mein St., Wellington 6021, New Zealand; liz.dennett@otago.ac.nz (E.R.D.); kirsty.danielson@otago.ac.nz (K.M.D.); 2Department of General Surgery, Wellington Hospital, 23 Mein St., Wellington 6021, New Zealand

**Keywords:** exosome, microvesicle, miRNA, liquid biopsy, non-coding RNA

## Abstract

Colorectal cancer (CRC) is one of the most common malignancies in the developed world, with global deaths expected to double in the next decade. Disease stage at diagnosis is the single greatest prognostic indicator for long-term survival. Unfortunately, early stage CRC is often asymptomatic and diagnosis frequently occurs at an advanced stage, where long-term survival can be as low as 14%. Circulating microRNAs encapsulated in extracellular vesicles (EVs) have recently come to prominence as novel diagnostic markers for cancer. EV-miRNAs are dysregulated in the circulation of CRC patients compared to healthy controls, and several specific miRNA candidates have been posited as diagnostic markers, including miR-21, miR-23a, miR-1246, and miR-92a. This review outlines the current landscape of EV-miRNAs as potential diagnostic markers for CRC, with a specific focus on those able to detect early stage disease.

## 1. Introduction

Colorectal cancer (CRC) is the third most frequently diagnosed cancer globally, with over a million patients diagnosed and half a million deaths annually [1]. By 2030, the worldwide incidence of CRC is expected to increase by up to 60%, resulting in 1.1 million deaths [2]. Early diagnosis of CRC is key for reducing CRC mortality; five-year survival for stage I and II disease is approximately 90%, which drops to 14% for stage IV disease [3]. Current screening and diagnostic strategies for CRC include faecal occult blood testing (FOBT), faecal immunochemical testing (FIT), flexible sigmoidoscopy, and colonoscopy. While these strategies work to reduce CRC mortality, issues with their use include limited sensitivity in early stage disease (faecal tests), invasiveness, and high resource burden [4]. The development of novel, sensitive, and minimally invasive diagnostic strategies for early stage CRC is essential to the continued reduction of CRC mortality.

MicroRNA (miRNA) are short, non-coding regulatory RNA molecules that are stable in the circulation, due in part to their encapsulation in extracellular vesicles (EVs). EV-miRNA are overexpressed in numerous cancers and have recently become the focus of an extensive amount of work aiming to identify novel biomarkers of disease [5,6,7]. This review will outline the current landscape of EV-miRNAs as potential diagnostic biomarkers for early stage CRC, as well as some of the current challenges to their clinical translation.

## 2. CRC Diagnosis and Screening

CRC is diagnosed following a symptomatic presentation, due to screening programmes, or as an incidental finding during investigations performed for unrelated reasons. Symptoms upon presentation can include weight loss, change in bowel habit, and perirectal bleeding; however, patients are often asymptomatic at early stages of disease [8]. Colonoscopy is the gold standard investigative and diagnostic tool in CRC. Unfortunately, extensive pre-procedure preparation, risk of complication, and high healthcare costs hinder its universal application for population-based screening in many countries [9].

Stool-based population screening programmes (FOBT and FIT) have been adopted in at least twelve countries [10]. FOBT is a guaiac-based test that detects occult blood in the stool, whereas FIT utilises immunoassays specific for human haemoglobin [11]. FIT is preferable over FOBT for population-based screening due to its higher sensitivity (79% vs. 71.2%) and comparable specificity (94% vs. 93.6%) for CRC detection [12,13,14,15]. A major limitation of faecal-based tests as a screening tool are their limited sensitivity for detection of adenomas and early stage CRC. FIT has been shown to have a higher percentage of false-negative results in detection of carcinoma in situ and T1 lesions when compared to more advanced stages, with a sensitivity as low as 66.7% for these early lesions [4].

While the adoption of population-based screening is a positive step toward reducing CRC-related mortality, screening uptake is not universal and is generally only available between the ages of 50–75 [10]. A considerable number of patients with CRC are either not eligible or do not participate in screening, likely only being diagnosed once the disease is more advanced [16]. Uptake rates of stool-based screening have been reported at 41–57%, and research suggests that there is a patient preference for blood tests compared to faecal-based screening [17,18,19]. A study by Osborne and colleagues found that 78% of participants would prefer to give a blood sample for CRC screening over a stool-based test [19].

## 3. Liquid Biopsy Biomarkers for CRC

The concept of a “liquid biopsy” stems from the discovery that cancer cells secrete molecules into the circulation that contain signature markers of their cell of origin [20,21]. The advantages of blood-based screening include minimal invasiveness, repeatability, and possible improved uptake compared to stool-based methods. Currently, no liquid biopsy markers are routinely used for screening or diagnosis of CRC. Carcinoembryonic antigen (CEA) is commonly used to monitor for recurrence; however, issues with sensitivity and specificity negate its use in diagnosis or screening [22].

The development of liquid biopsy biomarkers with higher sensitivity for early detection of CRC has the potential to dramatically improve rates of CRC-related mortality. Novel diagnostic and screening strategies may be used either in conjunction with, or as a replacement for, current stool-based screening programmes. Recently, various molecules have been identified as potential liquid biopsy markers. These include circulating tumour cells that originate from primary or metastatic sites, circulating cell-free tumour DNA (cfDNA), as well as miRNA, which can be free in the circulation or encapsulated in EVs [23,24].

Circulating cfDNA has been found to be significantly higher in CRC patients compared to healthy controls [25,26,27]. One form of cfDNA, methylated Septin9, has been approved by the United States (US) Food and Drug Administration as a PCR-based test for CRC screening. While early case-control studies of this test reported a 70% sensitivity and 90% specificity for CRC detection, testing in an asymptomatic cohort found only 35% sensitivity with 91% specificity for the detection of stage I CRC. This would hinder its viability for population-based screening for early stage cancers [28,29,30].

A significant amount of ongoing research is focused on the rapidly developing field of EVs. EVs are stable in the circulation and under various storage conditions [31,32], and reflect both the unique profile of the constituents of their cell of origin, as well as protecting these constituents in the circulation by encapsulation within a membrane. There has been increasing interest in examining the contents of these EVs, particularly miRNA, as candidates for liquid biopsy biomarkers.

## 4. Extracellular Vesicles (EVs)

EVs are a heterogeneous group of membrane-bound particles released from all human cells [33]. They have become of particular interest to cancer biomarker researchers due to the ability of tumour cells to secrete large amounts of EVs that contain protected tumour-specific cargo [34,35,36]. EVs also have a role as functional mediators of cancer cell biology. They can act in a paracrine fashion locally within the tumour microenvironment, and in an endocrine manner at distal sites via the circulation [37,38,39]. Pathophysiological functions of EVs in cancer include promotion of tumour invasiveness and growth [40,41], stimulation of stromal accessory cells to tumour-supporting phenotypes [42], and preparation of pre-metastatic niches in distant organs [43].

EVs are divided into three main categories based on biogenesis: Exosomes (~40–100 nm) are derived from multivesicular bodies within the cell’s endosomal system; microvesicles (or ectosomes/microparticles; 100 nm–1 µm) are formed from outward budding of the plasma membrane [44,45,46]; and apoptotic bodies (1–5 µm) arise from dying cells undergoing apoptosis [47,48,49]. In addition to these classes, some cancer-specific subtypes of EVs have been identified. Oncosomes (100–400 nm) are produced by non-transformed cells and their contents can produce oncogenic effects [50,51], whereas large oncosomes (1–10 µm) arise from malignant cells and are more atypical in morphology [52,53,54]. There is still significant disparity in the classification and analysis of EV subtyping in the literature, as well as inconsistencies in the isolation of “specific” EV subtypes. For this reason, we use the collective term EVs when referring to studies in this review.

EVs contain a range of cargo that reflect the parent cell of origin, including RNA, DNA, protein, and lipids [37,55]. DNA ranging in size from 100 base pairs to 2.5 kilobases has been identified within EVs [56]. Protein contents of EVs are related to both the route of biogenesis and the parent cell of origin. Commonly reported sub-classes of EV proteins include tetraspanins (e.g., CD53, CD63, CD81, and CD82) and accessory proteins essential to forming multivesicular bodies (Alix and TSG101) [44,47,57,58]. Surface proteins that may be enriched in CRC-derived EVs include A33, EpCAM, and CD147 [20,59]. Transfer of EV cargo between cells and subsequent functional effects have been demonstrated in numerous studies, highlighting the potential of EVs as both biomarkers and functional mediators of disease [46].

One of the most extensively studied classes of EV cargo to date has been miRNA, which are single-stranded, non-coding RNA molecules of approximately 18–22 nucleotides [60]. miRNA have been implicated in a host of normal biochemical processes, including cell differentiation, proliferation, and apoptosis [60,61,62,63]. miRNA levels are significantly dysregulated in CRC tumour tissue compared to normal colonic mucosa, and are thought to play a variety of roles in tumour development and biology [7,64,65]. In addition, accumulating evidence supports the existence of unique miRNA profiles in body fluids that may function as both diagnostic and prognostic biomarkers for cancer [66,67,68]. Circulating miRNAs remain stable in serum or plasma under a host of unfavourable conditions, including extremes of temperature and repeated freeze–thaw cycles [31,32], making them a highly appealing class of blood-based biomarkers. The stability of miRNAs in the circulation is due to their association with carrier molecules that protect them from degradation by RNAses [32]. This includes proteins such as Argonaute-2 and lipoprotein complexes, in addition to the aforementioned EVs [69,70].

## 5. EV-miRNAs as Diagnostic Biomarkers in CRC

There is some evidence that EV-specific miRNA may be of greater utility with regard to developing liquid biopsies in early stage CRC compared to total circulating miRNA levels. This is based on the concept that EV-miRNA profiles are more specific for tumour-derived signatures than total circulating miRNA due to the large volume of EVs released by tumour cells [6]. While sources of circulating EVs include platelets, red blood cells, and immune cells, in addition to tumour cells, CRC tumour cells release EVs in abundance in vitro [71,72]. Furthermore, EVs expressing specific surface markers can be isolated from the circulation to increase the specificity of tumour-derived EV biomarkers. One recent study has reported that the levels of CD147-expressing EVs are increased in the circulation of CRC patients versus controls as measured by flow cytometry [59]. Examples of other proposed enriched CRC-EV markers include EpCAM and A33 [20]. Interestingly, it also appears that cancer EVs contain higher levels of miRNA and can process pre-miRNA to mature miRNA [73]. This is a feature that is not present in EVs derived from normal cells, and may enhance the specificity of EV-miRNA as biomarkers of disease. Together, these data suggest that EV-miRNAs have significant potential as diagnostic biomarkers, due to being both robust in the circulation and storage in addition to their precision in reflecting the contents of their cell of origin.

## 6. Candidate EV-miRNAs for Use as Diagnostic Biomarkers

There have been numerous EV-miRNAs suggested as possible diagnostic markers in CRC. The most promising candidates that have been identified in one or more studies are outlined in Table 1 and Table 2. Research has also focused on using panels of EV-miRNAs, whereby measurement of multiple EV-miRNAs might offer improved diagnostic accuracy when combined, compared to any lone EV-miRNA. Studies have focused mainly on EV-miRNA derived from serum, plasma, or in vitro CRC cell lines.

It should be noted that EV-miRNAs have also been posited as useful prognostic markers in CRC. A number of EV-miRNAs are associated with progression of CRC or poorer overall survival. Specific examples include: Decreased expression of miR-4772 being significantly associated with recurrence of CRC [74] miR-27a and miR-130a being associated with poorer 5-year survival [75] and EV miR-30 being associated with metastatic progression of CRC [74]. For the purposes of this review, we focus on the use of EV-miRNAs in the diagnostic setting.

## 7. Diagnostic EV-miRNAs in Plasma and Serum

EV-miRNAs from plasma and serum that have been found to be dysregulated in CRC are listed in Table 1. There is significant variability with regard to the number of early stage patients included, which has the most relevance to applications as diagnostic or screening biomarkers. Ogata-Kawata and colleagues evaluated a cohort of 88 CRC patients, of which 20 each had stage I and stage II disease. They assessed the ability of various serum EV-miRNAs in differentiating these CRC patients from 11 controls. Their findings demonstrated that miR-23a, miR-1246, and miR-21 were able to differentiate CRC patients (all stages) from controls with an area under the curve (AUC) of 0.953, 0.948, and 0.798, respectively. A further validation cohort with a total of 7 stage I and 6 stage II patients vs. 8 controls demonstrated a significant difference in these EV-miRNAs between controls and stage I and II CRC combined (*p* < 0.001 for miR-1246; *p* < 0.0001 for miR-23a and miR-21) [76].

Serum EV miR-21 has similarly been found to be elevated in colonic adenoma, and can differentiate adenoma from controls with a sensitivity and specificity of 73.1% and 68.1%, respectively, corresponding to an AUC of 0.77 [21]. These data suggest that miR-21 could be elevated in serum EVs from early in the adenoma-carcinoma sequence and remains elevated in late stage disease, demonstrating obvious potential as a diagnostic biomarker [93]. Unfortunately, serum EV miR-21 does not appear to be exclusively dysregulated in CRC. It has also been observed to be elevated in hepatocellular, oesophageal, and breast cancers, as well as non-malignant conditions such as acute kidney injury [94,95,96,97]. This would be a barrier to its adoption as a population-screening tool as it may lack specificity for CRC. However, markers such as this may still have a role in population-based screening, in conjunction with existing methods. If employed following a positive FIT, miR-21 elevation might indicate a higher probability that CRC is present. It could then be used in combination with FIT to shorten time to wait for definitive diagnosis for these higher risk patients.

Yan and colleagues, found upregulation of miR-486 and downregulation of miR-548c when comparing serum EV-miRNA from 77 CRC patients, of which 26 were stage I and II, compared to 20 healthy controls [81]. Liu et al. likewise found elevated miR-486 levels in the serum EVs of CRC patients of all stages [74]. Downregulation of serum EV miR-548c was also reported by Peng and colleagues in 108 CRC patients (including 25 stage I and 21 stage II patients) and was associated with shorter survival and liver metastases [82]. However, in this study, miR-548c was demonstrated to have a 1.59-fold decrease and miR-486 was shown to have a 1.61-fold increase. While these differences are significant, their relatively low absolute change may lead to substantial overlap in levels between CRC and healthy patients, thus limiting their ability to differentiate between the two groups.

Matsumara et al. found that miR-1246 and miR-23a were elevated 2.23- and 2.7-fold, respectively, when comparing serum EV-miRNA from a group of 209 CRC patients, of which 107 were stage I and II, against 28 controls. Their findings also demonstrated a significant elevation of miR-92a, both in stage I and stage II patients (*p* < 0.05 for both) [77]. Chen et al. also described increased EV miR-1246 in 46 stage II patients compared to 50 controls (*p* < 0.0001) [6]. Yamada et al. again demonstrated elevation of plasma EV-derived miR-1246 and miR-92a in 13 mice bearing colon cancer xenografts [87,88]. Similarly, serum EV miR-92a is significantly elevated in patients with colonic adenoma and CRC when compared to controls [21,79]. In summary, miR-92a appears to be consistently elevated in both adenoma and early stage CRC [21,77,79].

## 8. EV-miRNAs in CRC Cell Lines and Tumour Tissue

To determine tumour-specific EV-miRNA markers, analysis was performed from EVs isolated from in vitro CRC cell line models (Table 2). There is some overlap between cell line-derived EV-miRNAs and the circulating EV-miRNAs in Table 1. Ji and colleagues used anti-EpCAM and anti-A33 antibodies to isolate EVs from CRC cell culture media [85]. They demonstrated elevation of miR-1246 and miR-23a in both A33- and EpCAM-positive EVs compared to the CRC cell lysate. These miRNAs are also elevated in serum-derived EVs of CRC patients [76]. Other studies have also found miR-1246 to be upregulated in EVs derived from a number of in vitro cell lines [78,86]. Additionally, Ji et al. found elevation of miR-19a and miR-203a in A33-positive EVs isolated from CRC cell line culture compared to CRC cell lysate. Both of these miRNAs have been similarly shown upregulated in serum EVs (see Table 1) [85].

miR-1246 has been observed to be both elevated and suppressed in CRC tumours compared to normal colonic tissue [78,98]. Scarpati and colleagues examined CRC tissue from 57 CRC patients, of which 24 were stage I and II, compared to normal stroma. Their results found miR-1246 to be upregulated relative to normal colonic tissue, with a 2.1-fold change (*p* < 0.0001) [98]. Yamada et al., conversely, demonstrated decreased expression of miR-1246 in tumour tissue compared to normal colonic tissue in 33 CRC patients, of which 13 were stage I and II [78]. Methodology differed in the processing of tissue samples between these studies. Scarpati and colleagues isolated tissue samples from formalin-fixed, paraffin-embedded (FFPE) slides, whereas Yamada et al. immediately froze the CRC tissue in liquid nitrogen [78,98]. This may be a possible reason for conflicting findings.

Yamada and colleagues also examined EVs isolated from CRC culture media from DLD-1, WiDr, SW480, and COLO201 cell lines [78]. Their findings demonstrated a significant enrichment in expression of miR-1246 within the EVs when compared to intracellular levels (*p* < 0.01) [78]. A further study also observed a similar pattern; miR-1246 was found to be elevated in CRC cell culture media EVs when isolated from the LM1863 cell line when compared to the cell lysate [85]. The findings of these studies suggest that miR-1246 may be packaged into EVs and subsequently released from CRC cells.

Similarly, miR-21 is upregulated in EVs isolated from CRC cell culture media, in addition to the circulation [87,88]. Upregulation of miR-200c from CRC cell line-derived EVs has also been found in at least four different studies; however, its presence in EVs in the peripheral circulation does not appear to have been established [85,87,89,90]. Although it has been demonstrated to be significantly upregulated in EVs isolated from mesenteric vein blood (*p* = 0.02) [99], it is possible that levels of miR-200c are altered by passing through the liver before joining the systemic circulation.

CRC cell culture models have also been used to examine the functional role of miRNAs. Whilst fully establishing the effect that miRNAs have on many biological processes is far from complete, there are links between some of the miRNAs outlined and oncogenic functions. This promotes the concept that using functional, rather than passive, biomarkers may be more useful when attempting to establish candidate molecules that can be adapted to a clinical setting. Functional biomarkers that have an active role in the disease process may be more likely to become dysregulated and change in tandem with progression of disease. Examples of miRNA biomarkers with oncogenic functions include miR-21, miR-1246, and miR-23a.

High expression of miR-21 in CRC cell culture has been associated with activation of the Wnt/β-catenin pathway that may promote tumour development, proliferation, and progression [100,101]. In addition, it is implicated in promoting the epithelial-to-mesenchymal transition, a key step in the formation of metastasis [102]. miR-1246 may have a role in promoting tumour progression. Yamada et al. demonstrated that miR-1246 is transported in EVs, which affects Smad signaling and modulates the tumour environment to promote angiogenesis and tumour growth [78]. miR-23a, which forms a cluster with miR-27a and miR-24-2, appears to be elevated in stage I and II CRC tissue, with a subsequent increase in miR-27a in more advanced cancers. The authors suggest that miR-23a promotes the migration and invasion of CRC cells, whereas miR-27a primarily promotes tumour proliferation [103].

## 9. Non-miRNA Non-Coding RNA

In addition to miRNAs, there are other types of non-coding RNA that are dysregulated in CRC and are detectable in EVs [104,105,106]. To date, these have not been explored to the same extent as miRNAs. There have been several recent studies that have begun to investigate their role in CRC; however, the expression of these candidate biomarkers in EVs in CRC remains to be examined. Examples of these molecules include long non-coding RNA (lncRNA) and circular RNA (circRNA); both have been found to be differentially expressed in EVs from CRC cell lines and may have potential as diagnostic biomarkers in CRC [107,108]. Graham et al. examined the lncRNA CRNDE and found it to be upregulated in CRC tumour tissue (*n* = 161) and adenoma (*n* = 29) compared to normal colon tissue from control patients (*n* = 222) [109]. A subsequent validation cohort of 20 CRC, 21 adenoma, and 30 control patients demonstrated an ability to discriminate adenoma and normal tissue, with a sensitivity and specificity of 95% and 96%, respectively, with an AUC of 0.938. Their study also looked at CRNDE levels in total plasma from 15 CRC and 15 control patients; this demonstrated a 5.5-fold increase in plasma from CRC patients and a subsequent AUC of 0.873 [109]. CCAT1, another lncRNA, has also been shown to be upregulated in CRC tumour tissue compared to matched normal mucosa, and adenoma compared to healthy controls [110]. Furthermore, a circRNA (circ_001988) has been shown to differentiate CRC tissue from normal mucosa in 31 CRC patients, of which 15 were stage I and II, with an AUC of 0.788 [111]. It also had a significant association with tumour differentiation and perineural invasion [111]. The role of CRNDE, CCAT1, and circ-001988 with respect to circulating EVs in CRC has not yet been examined.

Small nucleolar RNA (snoRNA) and PIWI-interacting RNA (piRNA) have also shown association with CRC. In a cohort of 188 CRC patients, high expression of SNORA42 in CRC tumours was associated with poorer overall survival, disease-free survival, and distant metastasis when compared to tumours with low expression; however, EVs were not assessed [112]. To date, one EV piRNA, piR-019825, has been examined. It was included with six miRNAs isolated from plasma EVs to form a panel that was effective in differentiating early stage CRC from healthy controls, discussed in Section 10 [84].

With regard to non-coding RNA in EVs, miRNAs have been the most extensively studied. It can be seen from the above research that there is also a significant amount of dysregulation in CRC with these other types of non-coding RNA. The current landscape for these novel molecules is rapidly developing. Similar types of non-coding RNA are known to be present in EVs in the circulation, and further research into their role as biomarkers in CRC with respect to EVs may be beneficial [104,105,106].

## 10. miR Panels

Combining a number of EV-miRNAs together in a panel is another option that is the focus of ongoing research. An obstacle in finding a single EV-miRNA that can function as a liquid biopsy marker is that most of the EV-miRNAs discussed have been implicated in multiple pathologies, both malignant and benign [113,114,115]. A further difficulty is that, by its nature, CRC tumorigenesis is not an organised and regulated process. There may be diversity of EV-miRNA profiles produced by CRC tumours. Due to this, a panel of EV-miRNAs have been employed in some studies to attempt to mitigate against this inherent heterogeneity.

Yuan et al. developed a six EV-miRNA panel that consisted of miR-1343, -125a, -708, -381, -543, and piR_019825 in 100 CRC patients (25 for each stage I–IV) and 50 controls. Their results demonstrated that both miR-125a and miR-1343 were significantly downregulated in the plasma EVs of all stages I–IV compared to controls (false discovery rate (FDR) < 0.05 for stage I vs. control and FDR < 0.05 for stage II vs. control for both miRNAs). Their six EV-miRNA panel was able to differentiate CRC from healthy controls with an AUC of 0.68, 0.77, 0.78, and 0.81 for stages I–IV, respectively. This was superior to any lone EV-miRNA, the the best performing of which were miR-125a that had an AUC of 0.62 and 0.76 for stages I and II, respectively, and miR-1343 which had an AUC of 0.59 and 0.72 for stages I and II, respectively [84].

Similarly, Ogata-Kawata and colleagues studied seven EV-miRNAs (let-7a, miR-21, -23a, -223, -150, -1229, -1246) which were all elevated in EVs from CRC patients’ serum and from CRC cell lines [76]. This study group included 88 CRC patients, of which 20 each were stage I and stage II, as well as 11 healthy controls. Combined use of this panel did not show increased diagnostic power compared to miR-1246 or miR-23a alone, which had AUC values of 0.948 and 0.953 for all stages of CRC, respectively [76].

## 11. Challenges in Using miRNAs as Diagnostic Biomarkers

There are a number of challenges in developing miRNA liquid biopsies in CRC. Much research has focused on total circulating, rather than EV-specific miRNA levels. Advantages of this include that there is no need to isolate EVs from which to extract miRNAs. This has efficiency benefits with regard to sample processing, but with potentially lower specificity [116].

Certain methods of EV isolation may not be easily scalable to the higher throughput of a hospital laboratory. Ultracentrifugation (UC), including density gradient and differential UC, is a commonly used method of EV isolation [6,20,76,77,79]. It is limited by the number of wells in the ultracentrifuge machine, usually six, in addition to taking up to 18 h to process samples [6]. Other methods of EV isolation, such as size exclusion chromatography (SEC) columns may be more easily translated to a clinical setting. These techniques are relatively rapid (approximately 15 min processing) and the number of samples that can be processed is only limited by the number of columns available. It is also possible to automate the process of isolating the EVs as they pass through the column [117]. This has possible efficiency benefits over a method such as UC, with potential of ease of scalability and automation. Other methods of EV isolation include precipitation and filtration that were also in a number of studies [74,80,81,82,118]. Differences in EV isolation methods, including UC, precipitation, and SEC, have demonstrated significant variance in the properties of the isolated vesicles, the total miRNA yield, as well as the EV-specific miRNA levels when analysing the same samples [119,120].

Other technical considerations that may contribute to conflicting results include the use of “housekeepers”, which are genes or miRNAs, which should not change in expression between controls and CRC. There was no consistency with the use of these housekeepers with multiple candidates chosen; examples include endogenous RNAs (e.g., RNU43), endogenous miRNAs (miR-16a, mir-451), and exogenous “spike in” controls (cel-miR-39-3p) [6,20,77]. This may introduce variance in the methods that might explain heterogeneity in the results. In addition, whether serum or plasma produce a higher miRNA yield is an area in which there is contradictory evidence. Contamination of a sample due to haemolysis, platelets, or other impurities may also contribute [121,122,123]. Wang et al. found that serum produced higher miRNA yields, whereas, conversely, McDonald and colleagues reported that plasma levels were higher [121,124]. Selection of control patients in this population can also be a contributing factor to inter-study variability. It is possible that “healthy controls” may have underlying adenoma or CRC, which can only be revealed via colonoscopy; up to 50% of patients under investigation for CRC by colonoscopy will have polyps [125]. The majority of studies did not state that all of their controls had a normal colonoscopy prior to recruitment.

There is a need to standardise methodology and develop technologies that make the isolation and detection of EVs feasible and scalable to the higher throughput of a hospital laboratory. Global adoption of the Minimal Information for Studies of Extracellular Vesicles (MISEV) 2018 guidelines across EV research may be a useful starting point to address some of these methodological issues [126].

It is necessary to evaluate how EV-miRNA diagnostic biomarkers can be integrated into a modern healthcare system. Ideally, a blood-based biomarker would outperform current faecal-based assays on all parameters for screening. However, isolating a single miRNA or miRNA panel that is both sensitive and specific to CRC may prove challenging. Using biomarkers in conjunction with current screening methods may provide an avenue for their initial adoption to clinical practice. Adopting a blood-based test following negative FIT in the context of ongoing symptoms suggestive of CRC is one option. Another possibility may be using blood-based markers to triage time to wait for colonoscopy. If there is a positive FIT in conjunction with a blood-based biomarker suggestive of CRC, investigations could be expedited to aid in earlier diagnosis and treatment.

## 12. Conclusions

EV-associated miRNAs appear to have great potential as biomarkers in CRC. Currently there are a number of challenges; standardisation of methodology may help with identifying candidate miRNAs that offer the greatest potential as liquid biopsies. Some of the results for circulating EV-miRNAs, including miR-23a, miR-1246, and miR-92a, are encouraging. A widespread issue with the field of miRNAs appears to be isolating markers that are specific to a certain disease, such as CRC. Most of the miRNAs discussed are dysregulated in a wide number of pathologies. Due to this, the concept of an miRNA panel has the potential to mitigate against some of these issues of specificity. Identifying a blood-based biomarker for CRC screening which outperforms faecal-based testing would be likely to significantly increase uptake in CRC screening. This, currently, is in the region of 50% of the eligible population. The subsequent effects would likely include earlier diagnosis of CRC and, thus, improved outcomes for the millions of people who will be affected by this disease.

## Figures and Tables

**Table 1 cancers-12-00052-t001:** Extracellular vesicle (EV)-micro RNAs (miRNAs) in plasma and serum.

miRNA	Source	Regulation ↑/↓	Sample Size (Stage I & II)	Findings	Ref.
**MIR-1246**	Serum	↑	88 CRC 11 HC (I 20, II 20)	CRC vs. HC AUC 0.948, true positive 95.5%, false positive 9%, Stage I vs. HC *p* < 0.001 (validation stage)	[76]
	Serum	↑	209 CRC 28 HC (I & II 107)	CRC vs. HC elevated 2.23-fold	[77]
	Plasma	↑	159 CRC 50 HC (I 58, II 46)	Stage II vs. HC *p* < 0.0001	[6]
	Plasma (Murine)	↑	13 xenograft mice 13 controls	Significantly elevated in 61% of samples	[78]
**MIR-21**	Serum	↑	88 CRC 11 HC (I 20, II 20)	CRC vs. HC AUC 0.798, true positive 61.4%, Stage I vs. HC *p* < 0.001 (validation stage)	[76]
	Serum	↑	26 adenoma 47 HC	Significantly elevated in adenoma vs. HC *p* = 0.0002, HC vs. Adenoma AUC 0.77	[21]
**MIR-23A**	Serum	↑	88 CRC 11 HC (I 20, II 20)	CRC vs. HC AUC 0.953, true positive 92%, false positive 0%, Stage I vs. HC *p* < 0.001 (validation stage)	[76]
	Serum	↑	209 CRC 28 HC (Combined I & II 107)	CRC vs. HC elevated 2.7-fold	[77]
	Plasma and Serum	↑	13 CRC 5 HC (I 1, II 4)	CRC vs. HC significantly elevated in EpCAM + EVs (*p* = 0.0075)	[20]
**MIR-92A**	Serum	↑	209 CRC 28 HC (I & II 107)	Stage I vs. HC *p* < 0.05, Stage II vs. HC *p* < 0.05	[77]
	Serum	↑	26 adenoma 47 HC	Significantly elevated in adenoma vs. HC *p* < 0.05	[21]
	Serum	↑	29 CRC 10 HC	CRC vs. HC AUC 0.845, significant correlation with T stage *p* < 0.001	[79]
	Plasma (Murine)	↑	13 xenograft mice 13 controls	Significantly elevated in 79% of samples	[80]
**MIR-548C-5P**	Serum	↓	77 CRC 20 HC	Significantly downregulated 1.59-fold *p* < 0.01	[81]
	Serum	↓	108 CRC (I 25, II 21)	Downregulation associated with shorter survival *p* = 0.18, liver metastasis, later stage	[82]
**MIR-486-5P**	Serum	↑	77 CRC 20 HC	Significantly upregulated 1.61-fold *p* < 0.01	[81]
	Serum	↑	84 CRC	Upregulated in exosomes in recurrent vs. non-recurrent patients—1.6- fold increase	[74]
**MIR-17**	Serum	↑	29 CRC 10 HC	CRC vs. HC AUC 0.897, significant correlation with T stage *p* < 0.001	[79]
**MIR-19A**	Serum	↑	209 CRC 28 HC (I & II 107)	Stage I vs. HC *p* < 0.001, Stage II vs. HC *p* < 0.001	[77]
**MIR-125A-3P**	Plasma	↑	50 CRC 50 HC (I 3, II 47)	Stage I and II vs. HC *p* = 0.0035, CRC vs. HC AUC 0.68	[83]
**MIR-125A-5P**	Plasma	↓	100 CRC 50 HC (I 25, II 25)	AUC 0.62, 0.76, 0.74, 0.77 for stages I–IV vs. HC respectively, overall AUC 0.75	[84]

HC: Healthy control; I: Stage I; II: Stage II; ↑: Upregulated; ↓: Downregulated; AUC: Area under the curve.

**Table 2 cancers-12-00052-t002:** EV-miRNAs from in-vitro colorectal cancer (CRC) cell lines.

miRNA	Source (CRC Cell Line)	Regulation ↑/↓	Findings	Ref.
**miR-1246**	LM1863	↑	Elevated in A33+ and EpCAM + exosomes	[85]
	HCT116	↑	Highly expressed in EVs from HCT116 cell line	[86]
	DLD-1, WiDr, SW480, COLO201	↑	Extracellular EV levels were significantly elevated compared to intracellular levels *p* < 0.01	[78]
**miR-21**	CaCo2, SW480, HT29	↑	Elevated in EVs in all cell lines	[87]
	SW480, WiDr	↑	Elevated in EVs in SW480, WiDr cell lines	[88]
**miR-23a**	LM1863	↑	Elevated in A33+ and EpCAM + EVs	[85]
**miR-200c**	LM1863	↑	Elevated in A33+ and EpCAM + EVs	[85]
	CaCo2, SW480, HT29	↑	Elevated in EVs in all cell lines	[87]
	SW480, SW620	↑	Present in EVs, increased in metastatic cell line following treatment with decitabine	[89]
	CCL227	↑	Upregulated in primary CRC, decreased level in EVs associated with increased invasiveness	[90]
**miR-203a**	LM1863	↑	Elevated in A33+ and EpCAM + EVs	[85]
**miR-145**	DLD-1	↑	Upregulated in EVs compared to intracellular levels	[91]
	DLD-1	↑	Higher EV levels in 5-FU resistant cells	[92]
**miR-17**	SW480, SW620	↑	Primary and metastatic cell line EVs both significantly upregulated compared to normal mucosa *p* < 0.01	[79]
**miR-19a**	LM1863	↑	Elevated in A33+ EVs	[85]
**miR-7641**	LM1863	↑	Elevated in A33+ and EpCAM+ EVs	[85]
	SW480, SW620	↑	Significantly enriched (log2FC 6.1–11.5) in both SW480 and SW620 EVs	[6]

HC: Healthy control; ↑: Upregulated; ↓: Downregulated; log2FC: log2 fold change.

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
