# Peer review of "Circulating Extracellular Vesicle MicroRNA as Diagnostic Biomarkers in Early Colorectal Cancer—A Review"

_cancers, 2019, doi:10.3390/cancers12010052_

Round 1

Reviewer 1 Report

The article is well written and understandable.

The title and conclusions are consistent with the data and arguments presented.

Actually, the topic about extracellular vesicle microRNA in early CRC is relevant and very interesting in translational cancer research. Finally, the article fits perfectly to the aim and scope of the journal.

However, in my opinion, I suggest some minor changes:

1. The abstract should be redrafted, because it is often vague and long-winded. I suggest to re-write it focusing on EV in CRC. 

2. EVs require a special "general" paragraph including EV-associated proteins and related biological functions.

3. EV microRNA should be separated  from the previous EV paragraph (see point 2).

4. Basic aspects of EV research are not fully clarified, and pathophysiological roles of EVs remain unclear. Please, add a short paragraph highlighting this issue.

Author Response

1. The abstract should be redrafted, because it is often vague and long-winded. I suggest to re-write it focusing on EV in CRC.

We thank the reviewer for this observation, the abstract has now been re-drafted to be more focused on EV miRNAs in CRC.

2. EVs require a special “general” paragraph including EV-associated proteins and related biological functions.

We have included a paragraph describing the general contents of EVs at the end of pg.6 of the manuscript, including different classes of protein that are commonly identified and the ability of EV cargo to transfer between cells. In addition, we have added information regarding the pathophysiological roles of EVs in cancer to the first paragraph of pg. 6.

3. EV microRNA should be separated from the previous EV paragraph (see point 2).

This section has been separated as suggested.

4. Basic aspects of EV research are not fully clarified, and pathophysiological roles of EVs remain unclear.

Please, add a short paragraph highlighting this issue. Please see response to point 2.

Reviewer 2 Report

The authors discussed the role of circulating EVs miRNAs in early CRC. The review is in line with the title and the topic is discussed appropriately.

However, some points should be better explained:

in the paragraph 1.2 relating to liquid biopsy biomarkers, the role of free circulating DNA should be discussed  In paragrapgh 1.3, the authors described the content of EV in terms of proteins, lipids and RNA. However, EVs could contain also DNA molecules. Please discuss also this point. In paragraph 1.5 and 1.6, the prognostic role of the different miRNAs should be better explained  

Author Response

1.In the paragraph 1.2 relating to liquid biopsy biomarkers, the role of free circulating DNA should be discussed.

We apologise for any lack of clarity around the description of cfDNA in this section. We have discussed methylated Septin9 which, to date, is the most advanced of this class of marker in CRC. We have amended this section to more clearly describe Septin9 as a cfDNA marker and included a generalised statement regarding the increased levels of cfDNA in CRC patients (pg. 5). While this is a highly interesting area of research, it could warrant a review entirely on its own and we have chosen to limit the length of discussion on this topic as our chosen focus is EV-miRNA for this particular manuscript.

2. In paragraph 1.3, the authors described the content of EV in terms of proteins, lipids and RNA. However, EVs could contain also DNA molecules. Please discuss also this point.

We have now included a comment on EV-contained DNA on pg. 6 of the manuscript.

3. In paragraph 1.5 and 1.6, the prognostic role of the different miRNAs should be better explained.

Thank you for your suggestion. We have included a paragraph acknowledging the prognostic role of EV-miRNA on pg. 8 of the manuscript and have stated that for the purposes of this review we will be focussing on the use of these markers in the diagnostic setting.

Reviewer 3 Report

In the review article 'Circulating extracellular vesicle microRNA as diagnostic biomarkers in early colorectal cancer – a review,' the author has outlined the current scope of extracellular vesicles as a potential diagnostic biomarker for early detection of colorectal cancer. However, there are some sections in the article that need to be revised for grammar (sentence formation). Also, the font, including its size, is not the same throughout the manuscript.
Lastly, in the section of EV-miRNA in CRC cell lines and tumor tissue, the author at one place discusses that levels of miR1246 are decreased in tumors while later it is said to increase in tumors (wrt reference 77).

Author Response

1. There are some sections in the article that need to be revised for grammar (sentence formation). Also, the font, including its size, is not the same throughout the manuscript.

We apologise for the inconsistencies in formatting and grammar.  We have reviewed the manuscript and made corrections throughout. We hope that it is now of a sufficient standard.

2. In the section of EV-miRNA in CRC cell lines and tumor tissue, the author at one place discusses the levels of miR1246 are decreased in tumors while later it is said to increase in tumors (wrt reference 77).

The section on miR-1246 has been rewritten to provide clarity to the discussion with respect to the levels of this miRNA in CRC (pg. 10-11).

Reviewer 4 Report

The current article reviews different potential biomarkers such as circulating extracellular vesicle microRNA for colon cancer screening. While colon cancer is one of the most common malignancies in the developed world, early detection, which is a key factor for survival, is frequently not possible due to limited sensitivity of current tests. That is why a number of molecules secreted by tumour cells into circulation have been tested for their potential screening value through a method called “liquid biopsy”. MicroRNA(miRNA) are short non-coding RNA molecules that are stable in the circulation, due in part to their encapsulation in extracellular vesicles (EVs). The current review analyses various studies focusing on the most promising biomarkers such as miR-21, miR-23a, miR-1246 and miR-92a. While present results vary between studies, the most common hindrance are methodological differences around EV isolation and sample analysis, which have manifested in conflicting results. Another challenge consists in the fact that isolating candidates that are specific to a certain disease, such as CRC, is difficult since most biomarkers appear to be related to more than one malignancy, thus defeating the purpose of an early sensitive screening method for colorectal cancer.

​The present review is written in a clear and concise manner, analysing data from literature with precision and clarity, proving to be a valuable asset in it’s field.

Author Response

There are no comments to address.